# Role of B Cell Lymphoma 2 in the Regulation of Liver Fibrosis in miR-122 Knockout Mice

**DOI:** 10.3390/biology9070157

**Published:** 2020-07-08

**Authors:** Kun-Yu Teng, Juan M. Barajas, Peng Hu, Samson T. Jacob, Kalpana Ghoshal

**Affiliations:** 1Molecular, Cellular and Developmental Biology Program, The Ohio State University, Columbus, OH 43210, USA; tky382002@gmail.com; 2Department of Pathology, The Ohio State University, Columbus, OH 43210, USA; JuanMartin.Barajas@STJUDE.ORG (J.M.B.); jacob.42@osu.edu (S.T.J.); 3Comprehensive Cancer Center, Wexner Medical Center, The Ohio State University, Columbus, OH 43210, USA; peng.hu@osumc.edu; 4Department of Cancer Biology and Genetics, The Ohio State University, Columbus, OH 43210, USA

**Keywords:** microRNA-122 (miR-122), liver fibrosis, hepatic stellate cell (HSC), BCL2, Venetoclax

## Abstract

MicroRNA-122 (miR-122) has been identified as a marker of various liver injuries, including hepatitis- virus-infection-, alcoholic-, and non-alcoholic steatohepatitis (NASH)-induced liver fibrosis. Here, we report that the extracellular miR-122 from hepatic cells can be delivered to hepatic stellate cells (HSCs) to modulate their proliferation and gene expression. Our published Argonaute crosslinking immunoprecipitation (Ago-CLIP) data identified several pro-fibrotic genes, including Ctgf, as miR-122 targets in mice livers. However, treating Ctgf as a therapeutic target failed to rescue the fibrosis developed in the miR-122 knockout livers. Alternatively, we compared the published datasets of human cirrhotic livers and miR-122 KO livers, which revealed upregulation of BCL2, suggesting its potential role in regulating fibrosis. Notably, ectopic miR-122 expression inhibited BCL2 expression in human HSC (LX-2) cells). Publicly available ChIP-seq data in human hepatocellular cancer (HepG2) cells and mice livers suggested miR-122 could regulate *BCL2* expression indirectly through c-MYC, which was confirmed by siRNA-mediated depletion of c-MYC in Hepatocellular Carcinoma (HCC) cell lines. Importantly, Venetoclax, a potent BCL2 inhibitor approved for the treatment of leukemia, showed promising anti-fibrotic effects in miR-122 knockout mice. Collectively, our data demonstrate that miR-122 suppresses liver fibrosis and implicates anti-fibrotic potential of Venetoclax.

## 1. Introduction

MicroRNAs are ~21–25 nucleotide non-coding RNAs that post-transcriptionally regulate gene expression in mammals [1,2]. MicroRNA-122 (miR-122) is a liver-specific and abundant microRNA that plays a critical role in lipid metabolism [3,4,5], cell differentiation [6], hepatic polyploidization [7], hepatitis B and C virus replication [8], acetaminophen toxicity [9], pulmonary inflammation [10], and hepatocyte innate immunity [11]. Reduced miR-122 expression has been consistently found in HCV-negative liver cancers and is correlated with metastasis in hepatocellular carcinoma (HCC) patients [12,13,14]. The role of miR-122 in mediating liver homeostasis is best demonstrated by genetic knockout mouse models, where mice livers lacking miR-122 develop spontaneous steatohepatitis, fibrosis, and eventually HCC [5,15].

Liver fibrosis is a process attributed to excessive accumulation of extracellular matrix (ECM) proteins due to repeated wound-healing responses, which occurs in most types of chronic liver diseases [16,17]. It is widely accepted that the hepatic stellate cells (HSCs) are the main collagen-producing cells during liver fibrosis [16,18,19]. When the liver is injured, damaged hepatocytes release TGFβ1 and ROS, while Kupffer cells release TNF-α and TGFβ1 to activate quiescent HSCs into myofibroblast-like cells [16,20,21,22,23]. In addition to HSCs, immune cells in the liver also contribute to fibrosis by secreting IL-6 and TNF-α to the microenvironment upon liver injury [24]. If the damage persists for an extended period of time, the dead hepatic cells are replaced by fibrotic tissue, which increases the stiffness of the liver and leads to cirrhosis [25,26]. Interestingly, the expression of miR-122 was drastically decreased in a mouse liver fibrosis model induced by carbon tetrachloride (CCl4) [27]. Several studies have shown that the circulating microRNAs could be used as biomarkers for various liver disease, including cirrhosis (reviewed in [28]).

BCL2, an apoptosis regulator localized to the outer mitochondrial membrane [29,30,31], plays an important role in liver fibrosis [32,33]. This unique protein is known to promote cell survival by inhibiting the action of pro-apoptotic proteins, such as BAX and BAK [34]. BCL2 and other pro-survival BCL2 family members, e.g., BCL-_X_L and MCL-1, inhibit BAX- and BAK- induced permeabilization of mitochondrial membrane and release of cytochrome C and ROS [35]. BCL2 controls cell survival and functions as an oncogene in different malignancies [31]. Overexpression of BCL2 in chronic lymphocytic leukemia (CLL) and acute lymphocytic leukemia (ALL) cells renders these cancer cells resistant to chemotherapy [36]. Similarly, overexpression of BCL2 in HSCs increases resistance of these cells to apoptotic stimuli, such as Fas activation and exposure to TNF-α [33]. Further, overexpression BCL2 in mice livers resulted in cholestatic liver fibrosis [32], indicating that BCL2 might be a pro-fibrogenic factor in the liver.

In this study, we explored the role of miR-122 in regulating liver fibrosis. We observed that the transport of extracellular miR-122 released from hepatocytes to HSCs resulted in inhibition of HSC proliferation and fibrotic gene expression. When miR-122 is depleted from the hepatocytes, pro-fibrotic genes were de-repressed in the HSCs. To further substantiate the role of miR-122 in regulating liver fibrosis, bioinformatic and molecular analysis were applied to identify pro-fibrotic factors that might be regulated by miR-122. We found that BCL2 expression was negatively regulated by miR-122 through c-MYC, which we previously showed to be indirectly suppressed by miR-122 through targeting the transcriptional activator (E2F1) and co-activator (TFDP2) in mouse hepatic cells [37]. Furthermore, Venetoclax, a FDA approved BCL2 inhibitor for leukemia, suppressed proliferation of a human HSC cell line and expression of fibrotic genes. Venetoclax also reduced the collagen level and expression of fibrotic genes in the mice livers depleted of miR-122. These observations provide us a strong rationale to block liver fibrosis by targeting BCL2.

## 2. Materials and Methods

### 2.1. Dataset Analysis

All the dataset analysis was done using R software (version 3.4.0). The miR-122 expression data in cirrhotic patients (risk factors: Hepatitis B (HBV) and Hepatitis C (HCV)) were queried from ArrayExpress (https://www.ebi.ac.uk/arrayexpress/) (Accession number # E-TABM-866) [38,39] hosted by The European Bioinformatics Institute (EMBL-EBI). Expression profiles of miR-122 KO mouse were downloaded from Gene Expression Omnibus (GEO) (https://www.ncbi.nlm.nih.gov/geo/) using GEOquery package from Bioconductor (https://bioconductor.org/). GSE20610 (microarray of wild type (WT) and miR-122 liver specific knockout (LKO) mice) [5] and GSE97060 (RNA-seq of WT and miR-122 knockout (KO) mice) [40] were used to identify potential fibrotic genes regulated by miR-122. BCL2 mRNA expression was extracted from GSE84044 (HBV-induced cirrhosis) [41] and GSE14323 (HCV-induced cirrhosis) datasets [42]. Argonaute crosslinking immunoprecipitation (Ago-CLIP) data were retrieved from GEO (GSE97058). Chromatin Immunoprecipitation—sequencing (ChIP-seq) data (BED file) were retrieved from Cistrome (http://cistrome.org/) and analyzed by UCSC genome browser (https://genome.ucsc.edu/).

### 2.2. Mouse Husbandry and Treatment

miR-122 germ-line knockout (KO) and liver-specific knockout (LKO) mice were generated as described [5]. All animals were housed in temperature-controlled and Helicobacter-free facility under a 12/12-h light/dark cycle. Animals studies were conducted under the guidelines of the Ohio State University Institutional Laboratory Animal Care Committee.

Carbon tetrachloride (CCl4) (Sigma-Aldrich, # 289116) was dissolved in corn oil and administered to miR-122 KO and wild-type (miR-122^fl/fl^) (WT) mice by intraperitoneal (i.p) injection. WT and miR-122 KO mice (~4 months old, 25 g) were treated with vehicle (corn oil) or CCl4 (0.25 mL/ kg) twice a week for 4 weeks. Mice were harvested one day after the last injection.

Venetoclax (ABT-199) was purchased from PharmaBlockUSA Inc. (cat# PBLJ6036) (Fisher Scientific). Venetoclax stock (25 mg/mL) was dissolved in dimethyl sulfoxide (DMSO), which was diluted in 10% 2-hydroxypropyl-β-cyclodextrin (HPBCD, cat# THPB-P, CTD, Inc) to reach final concentration at 2.5 mg/mL immediately before treatment of mice. Venetoclax (15 mg/kg) was administered to miR-122 KO mice that developed hepatic fibrosis at ~8 months of age [5] by i.p injection for 8 total doses in 4 weeks. 

### 2.3. Histological Analysis

Mice livers were fixed in 4% paraformaldehyde for 24 h then switched to PBS before embedding into paraffin. Paraffin embedded tissue sections (4 mm) on glass slides were subjected to H&E staining, Sirius red staining and α-smooth muscle actin (α-SMA) immunohistochemistry (IHC) as previously described [43].

### 2.4. Proliferation Assay

LX-2, a human hepatic stellate cell line, was maintained in DMEM supplemented with 10% FBS [44]. Cell proliferation was measured by a CellTiter-Glo^®^ Luminescent Cell Viability Assay kit (Promega, #G7572) following the manufacturer’s protocol.

### 2.5. Luciferase Assay

Mouse *Ccn2* gene (that codes for Ctgf) exon 1 harboring its 5′UTR was cloned into psi-Check2 at the Nhe I site located in the 5′UTR of the Renilla luciferase reporter. The inserted sequences and orientations were confirmed by Sanger sequencing. The psi-Check2 harboring the mutant miR-122 seed sequence in Ctgf 5′UTR was obtained using overlapping PCR. Primers used to generate wild type (WT) and mutant Ctgf 5′UTR are provided in Appendix A.

Recombinant psi-Check2 harboring Ctgf WT or mutant plasmids (50 ng) were co-transfected with either the scramble RNA or miR-122 mimic (25 nM) into mouse Hepa1-6 cells using lipofectamine 2000 (Thermo, #11668019) following manufacturer’s instructions. After 48 h, luciferase activity was measured using the Dual-Luciferase^®^ Reporter Assay System (Promega, #E1980) following manufacturer’s instructions.

### 2.6. Co-Culture of HCC Cells and LX-2 Cells

HCC cell lines HCCLM3 and PLC/PRF5 were engineered to overexpress miR-122 using lentivirus as described in [45]. Co-cultures were performed in a transwell plate (6.5 mm, 3µm pore polycarbonate membrane insert, Costar, # 3415). Briefly, HCC cells (10^5^ cells) were seeded in the top chambers, while LX-2 cells (2 × 10^5^ cells) were seeded in the bottom wells. Cells were cultured in the DMEM supplemented with 10% exosome-depleted FBS (SBI, # EXO-FBS-50A-1). After 72 h, culture supernatants were collected and exosomes and microvesicles were purified using the ExoQuick TC (SBI, # EXOTC10A-1) kit following the manufacturer’s guidelines. Purified microvesicles and co-cultured LX-2 cells (recipient) were subjected to the RNA isolation and RT-qPCR analysis described below.

### 2.7. Reverse-Transcription Polymerase Chain Reaction (RT-qPCR)

Total RNA was isolated by using TRIzol (Life Technologies, # 15596018) and purified following DNase I digestion. DNA-free RNA was reverse-transcribed into complementary DNA (cDNA) using a high-capacity cDNA reverse transcription kit (Applied Biosystem, # 4368813). RT-qPCR analysis of each sample was performed in triplicate using the SYBR Green method. Relative gene expression was calculated using the delta-delta-CT (ΔΔCT) method with the level *of ACTB* gene as the normalizer. Primer sequences are provided in Appendix A. To detect the primary transcripts and mature miR-122, TaqMan assays were used (Applied Biosystems, #Hs03303072 for primary miR-122; #TM: 002245 for mature miR-122). The expression of the primary and mature miR-122 transcripts was calculated by ΔΔCT method using RNU6B (Applied Biosystems, #TM: 001093) as the normalizer. 

### 2.8. Immunoblot Analysis

Proteins were extracted from whole cells or liver tissues in RIPA buffer followed by immunoblotting with primary antibodies as described in [43]. Near-infrared (NIR) fluorescence signals were developed using an Odyssey^®^ CLx Imaging System (LICOR). Primary antibody information is provided in Appendix A.

### 2.9. Statistical Analysis

Statistical analysis for patient datasets was performed with R (3.4.0). Expression counts of each sample per gene were log2-transformed. Significance was determined using the Wilcox.test() function in R. All bar diagrams in this study were shown as the mean ± standard deviation. Two sample *t*-tests were used for two-group comparisons. For more than two-group comparisons, ANOVA was performed. The *p*-values < 0.05 were considered significant and are represented with asterisks in the figures.

## 3. Results

### 3.1. miR-122 Is a Negative Regulator of Liver Fibrosis

Studies from different groups including ours have demonstrated development of spontaneous liver fibrosis upon depletion of miR-122 in mice livers [5,15], suggesting that miR-122 is an anti-fibrotic microRNA in the liver. To confirm that this association was common between mice and humans, the expression profiles of miR-122 in the normal livers and HCC-adjacent benign cirrhotic tissue was evaluated by analyzing the microRNA microarray data (E-TABM-866) downloaded from the European Bioinformatics Institute (EMBL-EBI) [39]. The miR-122 expression was evaluated in a total of 111 samples (21 normal liver; 90 cirrhotic livers). Significantly reduced levels of miR-122 (*p* < 0.05) were found in the cirrhosis patient group (Figure 1A). This data, along with the phenotype of the miR-122 KO mice [5], suggest that miR-122 might function as an anti-fibrotic molecule in the liver. To confirm the potential anti-fibrotic role of miR-122, LX-2 cells (an immortalized human hepatic stellate cell line) [44] were transfected with either the scrambled RNA (negative control) or miR-122 mimic, and the expression of miR-122 was confirmed by RT-qPCR (Appendix A). After 72 h, miR-122 transfected LX-2 cells showed reduced proliferation compared to the control group (Figure 1B). This data correlated with reduced expression of fibrotic genes such as *ACTA2* and *COL1A1* in the miR-122 transfected LX-2 cells (Figure 1C). Additionally, mice lacking hepatic miR-122 were found to be sensitive to the fibrosis stimulus (i.e., carbon tetrachloride (CCl4), 0.25 mL/kg), as demonstrated by a ~3.5-fold increase in the hydroxyproline level, a measure of collagen content (Figure 1D), the elevated collagen deposition as visualized by Sirius red staining, and activation of hepatic stellate cells as detected by the α-SMA immunohistochemistry in the livers of miR-122 KO mice (Figure 1E). These data indicate an anti-fibrotic role of miR-122 in the liver.

### 3.2. Extracellular miR-122 Released from Hepatic Cells Modulates Fibrotic Gene Expression in HSCs

It is well established that hepatic stellate cell (HSC) activation is a key event during liver fibrosis [16,46], and loss of miR-122 in the hepatocytes leads to spontaneous liver fibrosis [5,15]. We, therefore, sought to find out how hepatocyte-generated miR-122 could regulate HSCs. It has been experimentally shown that miR-122 could be transferred from cells abundantly expressing miR-122 to non-expressing cells [47], and the circulating extracellular vesicles (e.g., exosomes) were found to carry miR-122 [48,49]. To test whether hepatocyte-derived miR-122 could be delivered to HSCs to regulate their functions, two HCC cell lines (PLC/PRF5 and HCC-LM3) expressing miR-122 were co-cultured with LX-2 cells using a transwell system (Figure 1F). The expression of miR-122 was higher in the extracellular vesicles harvested from supernatants of co-cultured cells, as well as the recipient LX-2 cells co-cultured with miR-122 expressing donor HCC cells (Figure 1G,H). In addition, the increased miR-122 level in the recipient LX-2 cells showed a negative correlation with the expression of *COL1A1* and *ACTA2* (Figure 1H), suggesting that the extracellular miR-122 might modulate fibrotic gene expression in the HSCs. We also measured the expression of the primary miR-122 transcript (pri-miR-122) in all the recipient LX-2 cells using a TaqMan assay. Interestingly, only the positive control cells (Huh7) but not LX-2 cells exhibited pri-miR-122 expression (Appendix A), indicating that miR-122 in the co-cultured LX-2 cells was exogenously provided by the donor HCC cells.

### 3.3. Ago-CLIP Identified Several Profibrotic Targets of Mir-122 

Argonaute crosslinking immunoprecipitation (Ago-CLIP) is known for its ability to determine the precise sequences of the miRNA-mRNA interactions biochemically [8,50]. To look for the potential fibrotic genes that are responsible for the liver fibrosis induced by the loss of miR-122, we revisited the miR-122 target list generated by Ago-CLIP performed in our earlier study using WT and KO mice [40]. Several pro-fibrotic transcripts were found to harbor the miR-122-dependent peaks in the wild-type but not in KO livers, indicating miR-122 binding to these transcripts (Figure 2A,B and Table 1). Among these profibrogenic factors, KLF6 [15], P4HA1 [27] and TGBFR1 [51] have already been reported as miR-122 targets. Notably, we found that *Ctgf* mRNA harbors one miR-122 binding sites (7merA1) at its 5′ UTR. This is a rare binding event, as most of the miRNA and mRNA interactions are located at the 3′UTRs or coding sequences (CDS) of the targets. Due to this unique binding and well-characterized pro-fibrotic features of CTGF, we queried the relationship between CTGF and miR-122. An inverse correlation between miR-122 and *Ctgf* expressions was found by transfecting the miR-122 mimic in the KO hepatocytes or transfecting antisense miR-122 in the WT hepatocytes (Figure 2C). Furthermore, we validated the binding of miR-122 at the 5′UTR of mouse *Ctgf* mRNA by inserting the 5′UTR of *Ctgf* into the 5′UTR of the *Renilla* luciferase (rluc) reporter in psi-Check2 vector. A small (~25%) but significant decrease in rluc activity was observed in cells co-transfected with miR-122 mimic and psi-check2 harboring wild-type *Ctgf* - 5′UTR (Ctgf WT), which was abrogated in cells co-transfected with the miR-122 cognate site mutant (Ctgf mutant) vector (Figure 2D,E). After confirming that CTGF is a bona fide miR-122 target, we attempted to inhibit liver fibrosis developed in the miR-122 KO mouse by blocking the CTGF function; however, we did not find a significant improvement (data not shown).

### 3.4. Pro-Fibrotic BCL2 Is Overexpressed in miR-122 KO Mice and Human Cirrhotic Livers

Previously we [5] and others [15] have reported that miR-122 depleted mice develop spontaneous liver fibrosis with age. Therefore, we hypothesized pro-fibrotic factors that are highly expressed in the miR-122 KO mice might serve as therapeutic targets for liver fibrosis. To this end, we looked for fibrotic genes that are commonly dysregulated in both miR-122 KO mice and human cirrhotic livers by analyzing the RNA-seq and microarray data in the wild type and miR-122-depleted mice [5,40] and microarray data sets from the patients [41,42]. We specifically focused on pro-fibrotic genes that are upregulated in the KO mice and cirrhotic livers, which might be associated with the loss of miR-122 function. Among 12 fibrotic genes common in both (Figure 3A, Appendix A), we were particularly interested in B-cell lymphoma 2 (BCL2), an anti-apoptotic and pro-fibrotic gene reported in several studies [32,33,52]. Analysis of the top 100 overexpressed genes in the cirrhotic patients revealed that BCL2 served as a hub gene (a highly connected node in the protein interaction network) (Appendix A). Notably, BCL2 expression was upregulated in both HBV- and HCV-infected livers of cirrhosis patients (Figure 3B,C) [41,42]. More importantly, we found a significant negative correlation between miR-122 and BCL2 expression in human fibrotic livers (Figure 3D). Notably, increases in hepatic *Bcl2* mRNA in the miR-122 germline (KO) and liver-specific miR-122 knockout (LKO) mice compared to WT mice (Figure 3E,F) correlated with the de-repression of *Col1a1* and *Acta2* genes (Appendix A). Immunoblot analysis showed increased BCL2 protein levels in KO liver extracts (Figure 3G). Collectively, these data suggested that BCL2 might be involved in promoting fibrosis in miR-122-depleted livers.

### 3.5. miR-122 Regulates BCL2 Expression via c-MYC

To verify the impact of miR-122 on *BCL2* expression, miR-122 mimic or scrambled mimic RNA was transfected into immortalized human hepatic stellate (LX-2) cells and BCL2 mRNA level was assessed by RT-qPCR. Results revealed downregulation of BCL2 mRNA by ectopic expression of miR-122 (Figure 4A). Given that miR-122 could affect *BCL2* expression in both human cells and mice livers, we postulated that *BCL2* might harbor a miR-122 seed region. Unfortunately, we did not find any miR-122 seed sequence in the *BCL2* transcript, either by TargetScan database search (http://www.targetscan.org/) or in our Ago-CLIP data in mouse and human livers [40]. Hence, we hypothesized that miR-122 might regulate *BCL2* expression indirectly at the transcriptional level. Indeed, we found the primary transcript of *BCL2* (hnRNA) was downregulated upon ectopic expression of miR-122 in LX-2 cells (Figure 4B). To look for a potential transcription factor that could be regulated by miR-122 to modulate *BCL2* transcription, we queried the rVista database (https://rvista.dcode.org/). Among six potential transcription factors (conserved between mice and humans) that bind at the proximal promoter region of the identified *BCL2* gene (Appendix A), we focused on c-MYC as it is known to modulate liver fibrosis [53] and is regulated by miR-122 through E2F1 and TFDP2 in mouse hepatic cells [37]. Notably, ectopic miR-122 expression also inhibited c-MYC expression in LX-2 cells (Appendix A). We then searched Cistrome (http://cistrome.org/) database to find out if c-MYC binds to the BCL2 promoter. ChIP-seq data revealed strong c-MYC binding at a site in the BCL2 proximal promoter region in both the mice livers and human HepG2 cells (Figure 4C,D). Furthermore, the c-MYC binding profile overlapped with RNA polymerase II (POL II) and histone H3 trimethyl lysine 4 (H3K4me3) peaks, which are markers of active genes (Figure 4C,D), suggesting active *BCL2* transcription upon c-MYC binding. We next sought to determine whether the BCL2 expression is modulated by c-MYC. To this end, we used siRNA to knockdown c-MYC in the LX-2 cells. BCL2 protein as well as the primary transcript (hnRNA) and mRNA levels were reduced in LX-2 cells after siRNA-mediated c-MYC depletion (Figure 4E,F), indicating that c-MYC positively regulates *BCL2* transcription. We further verified the relationship between c-MYC and BCL2 using a murine HCC cell line (EC4) that induces c-MYC expression upon doxycycline withdrawal [54]. This cell line was derived from the liver tumors spontaneously developed after induction of c-MYC in hepatocytes upon doxycycline withdrawal in the double transgenic (LAP-tTA; Tet-O-Myc) mice [54]. We found that the BCL2 protein level was reduced upon c-MYC suppression in EC4 cells (Appendix A). Collectively, these data, along with our previous findings [37], suggest that miR-122 regulates BCL2 expression in the liver, at least in part, through c-MYC.

### 3.6. BCL2 Blockade Reduces Proliferation of Hepatic Stellate Cells (HSCs) and Suppresses Liver Fibrosis in miR-122 KO Mice

To study the role of BCL2 in HSC activation and fibrosis, we used Venetoclax, a selective inhibitor approved by the Food and Drug Administration (FDA) for chronic lymphocytic leukemia patients with chromosome 17 p deletion [55]. We found a dose-dependent sensitivity of LX-2 cells to Venetoclax (Figure 5A). Venetoclax treatment inhibited 50% of cell proliferation (IC_50_) at ~10 µM concentration in LX-2 cells after 72 h (Figure 5B). Proliferation of these cells was significantly reduced with time upon this drug treatment (Appendix A). It has been reported that Venetoclax could interrupt the interaction between BCL2 and the BIM complex, thereby destabilizing the BIM protein and activating PARP cleavage in a BCL2 expressing the small-cell lung cancer cell line [56]. We postulated that Venetoclax would have a similar impact on the LX-2 cells [33]. Indeed, we found that Venetoclax treatment reduced the BIM level and activated PARP cleavage in LX-2 cells (Appendix A), suggesting disruption of the BCL2–BIM complex by the drug. Additionally, Venetoclax inhibited the expression of pro-fibrotic *COL1A1* and *ACTA2* genes in LX-2 cells (Appendix A). Together, our in vitro data suggest Venetoclax could serve as a potent inhibitor of hepatic stellate cell activation. 

As a proof of concept, we next investigated whether Venetoclax could block fibrosis in vivo by treating ~32-week-old miR-122 KO mice with the vehicle or Venetoclax (Figure 5C,D). These mice developed liver fibrosis by 24 weeks [5]. During treatment, no obvious changes in their body weights were observed (Appendix A). H&E and Sirius red staining showed that both collagen levels and immune cell infiltration were significantly suppressed in Venetoclax-treated mice compared to the vehicle-treated mice (Figure 5E,F and Appendix A). The expressions of the fibrotic genes (*Col1a1 and Acta2*) were downregulated in the Venetoclax-treated group compared to the vehicle controls (Figure 5G). Consistent with the histological evidence, Venetoclax also reduced the hydroxyproline content in the miR-122 KO livers by ~50% (Figure 5H), underscoring the significance of BCL2 in mediating liver fibrosis and the therapeutic efficacy of Venetoclax in vivo.

A mouse liver section stained with Sirius red was blindly evaluated and the fibrotic score (mean ± SD) was given as follows: 1: no to low fibrosis; 2: mild fibrosis; 3: severe fibrosis (Table 2).

## 4. Discussion

In our earlier study, we reported that miR-122 LKO and KO mice develop spontaneous liver fibrosis [5]. Although a few studies have demonstrated that de-repressed miR-122 targets, e.g., *KLF6, P4HA1 and TGFBR1* could promote liver fibrosis [15,27,51], the mechanisms of how hepatocyte—produced miR-122 could regulate HSC and whether these miR-122 downstream molecules could serve as a therapeutic target for liver fibrosis are still not clear. In the present study, we focused on elucidating the mechanistic role of miR-122 in blocking fibrosis and identification of the potential therapeutic targets for liver fibrosis. To elucidate the connections between predominantly hepatocyte-expressed miR-122 and stellate cells, we used a transwell system to co-culture miR-122-expressing HCC cells with non-expressing HSC (LX-2) cells. As with other microRNAs that could be transferred between hepatocytes and HSCs [57], our data showed that miR-122 could be delivered from hepatic cancer cells to stellate cells through extracellular mechanisms. This notion is further supported by the enrichment of miR-122 in extracellular vesicles purified from the co-cultured media of miR-122 expressing HCC cells and LX-2 cells (Figure 1G). The increased level of mature miR-122 in HSCs, but not the primary miR-122 transcripts, suggests that the source of miR-122 is exogenous (from HCC donor cells). Furthermore, we showed that the elevated miR-122 level was inversely correlated with the reduced expression of fibrosis genes in LX-2 cells, indicating that the increased uptake of miR-122 in the receipt cells is functional. Supported by another study [27], our data not only showed that miR-122 is an anti-fibrotic microRNA, but also demonstrated that miR-122 could be delivered from hepatic cells to HSCs to inhibit cell proliferation and expression of fibrosis genes (Figure 6). 

To explore the downstream molecules that are responsible for liver fibrosis developed in miR-122 KO mice, we analyzed the Ago-CLIP data in the miR-122 KO mice. We identified several of pro-fibrotic factors that are targeted and regulated by miR-122 (Table 1). We specifically focused on CTGF, as miR-122 has a unique binding site located at its 5′UTR. More importantly, unlike other well-known fibrotic factors such as TGFβ and PDGFβ [58,59], the anti-fibrotic role of CTGF in the liver is yet to be established. We, therefore, did a series of experiments to prove that miR-122 indeed regulates CTGF. Moving forward, we aim to suppress liver fibrosis developed in miR-122 KO livers by blocking CTGF functions. Admittedly, we did not find significant improvements in these mice by blocking CTGF function in vivo. It was evident that focusing only on the mouse miR-122 pro-fibrotic targets may not be adequate to identify a druggable fibrotic factor. This observation prompted us to combine both mouse and human gene expression data collected from the normal and fibrotic livers, which showed dysregulation of 12 fibrotic factors (Figure 3A, Appendix A). Before further validation, it was imperative to identify a fibrosis hub gene that is regulated by miR-122 in the liver. BCL2 stood out as the potential candidate that fulfilled the established criteria for its role in fibrosis. However, miR-122 does not appear to regulate BCL2 expression by direct binding to its mRNA because Ago-CLIP data did not identify a direct interaction of the two [40], and no miR-122 seed sequence was identified in *BCL2* mRNA. Interestingly, we found that miR-122 could regulate *BCL2* transcription through c-MYC. However, the role of c-MYC in *BCL2* gene regulation is complex; it depends on the cellular milieu, such as the cell type, oncogenic status, c-MYC level, and p53 status [60,61]. 

It is noteworthy that in addition to its role in liver fibrosis, BCL2 is known to play a critical role in leukemia [31,62,63]. The anti-apoptotic functions of BCL2 help tumor cells overcome apoptosis stimuli, such as radiation. Many studies have been exploring the potential of targeting BCL2 to suppress tumor growth [31,34]. Although the first generation BCL2 inhibitor Navitoclax (ABT-263), had profound effects in suppressing leukemia, metastatic lung cancer, and senescent cells [64,65,66], it also targets BCL-X_L_, another BCL2 family member. These off-target effects caused thrombocytopenia in patients with relapsed or refractory chronic lymphocytic leukemia (CLL) [67,68]. The second-generation BCL2 selective inhibitor Venetoclax (ABT-199) did not cause this problem due to its high specificity toward BCL2 [69,70]. Venetoclax was approved by the FDA in June 2018 (https://www.fda.gov/) for therapy in CLL patients with a chromosomal deletion at 17p, which probably causes general resistance to chemotherapy [71,72]. Moreover, Venetoclax has been used in combination with rituximab to treat patients with CLL or small lymphocytic lymphoma (SLL). Thus, it is reasonable to use Venetoclax as a tool to verify the mechanistic role and to test the therapeutic value of targeting BCL2 for liver fibrosis. Indeed, we found that Venetoclax suppressed cell proliferation and fibrogenic gene expression in the human-liver-derived stellate cell line (LX-2). Venetoclax also suppressed liver fibrosis developed in the miR-122 KO mice, as demonstrated by the reduction in fibrosis marker expression, collagen deposition, and the hydroxyproline content (a marker of collagen level) in the liver. These data, along with the dysregulation of *BCL2* in both cirrhosis patients and mouse fibrosis models, suggest its potential for treating liver fibrosis.

Finally, drug re-purposing, or use of existing drugs for new diseases, has gained popularity due to the low cost and time-saving benefits in clinics. Compared to other anti-fibrotic drugs that are still in early clinical trials [73], our study identified an FDA-approved drug originally designed for CLL patients, for treatment of liver fibrosis. We selected BCL2 as a therapeutic target based on its expression profiles in two mice models and in human cirrhotic livers from published datasets deposited in GEO and EMBL-EBI. Our analyses clearly demonstrate that *BCL2* is upregulated in both human and mouse fibrotic livers. Inhibition of BCL2 functions using Venetoclax suppressed human HSC-derived cells (LX-2) from transforming into pro-fibrogenic status and inhibited liver fibrosis that developed in the miR-122 KO mice. With the mechanistic and phenotypic data generated in the current study, it is possible that blocking BCL2 using Venetoclax might be able to serve as a therapeutic agent for cirrhosis patients with high BCL2 expression.

## 5. Conclusions

Our Ago-CLIP analysis biochemically identified several novel profibrogenic targets of miR-122 that are upregulated in miR-122 livers. Among these, CTGF was validated as a unique 5’-UTR - specific miR-122 target. However, blocking CTGF did not resolve liver fibrosis in miR-122 knockout mouse model. Notably, bioinformatic analysis identified BCL2 as a major Hub gene, which is upregulated in human cirrhotic livers and its expression inversely correlated with the miR-122 level. We also found that anti-apoptotic and fibrogenic BCL2 is transcriptionally upregulated through c-Myc in miR-122- depleted livers and hepatic stellate cells. Furthermore, our study revealed that Venetoclax, a BCL2 inhibitor, approved by FDA for chronic lymphocytic leukemia, inhibited stellate cell proliferation and activation in vitro and impeded liver fibrosis in miR-122 knockout mice. Thus, the current study provides the rational and impetus for testing therapeutic efficacy of Venetoclax in different models before its clinical trials in patients with liver fibrosis. 

## Figures and Tables

**Figure 1 biology-09-00157-f001:**
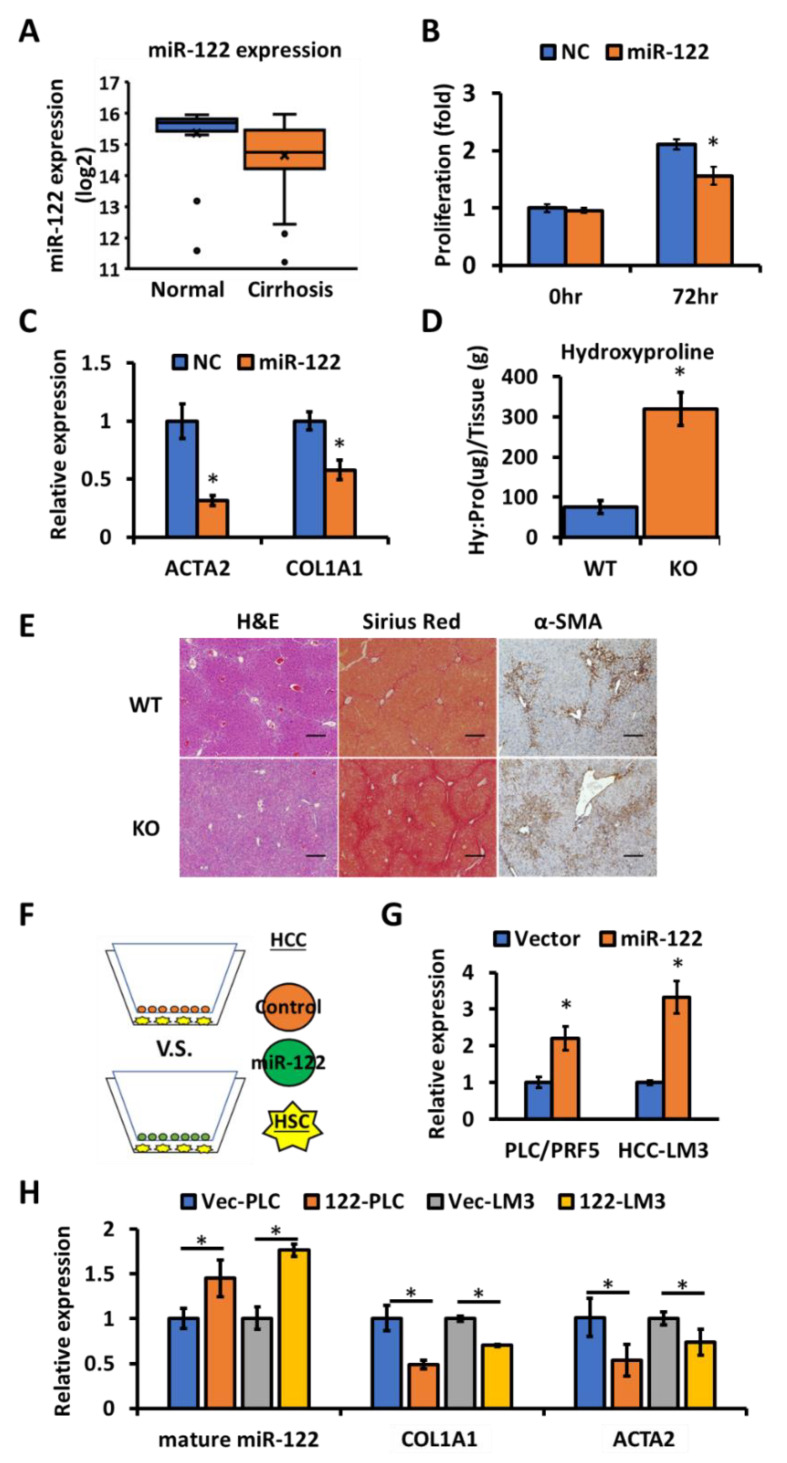
Anti-fibrotic role of miR-122 in both mouse and human fibrotic livers, and delivery of functional miR-122 into LX-2 cells through extracellular vesicles. (**A**) The miR-122 expression in the normal liver (*n* = 21) and non-tumor cirrhotic liver tissues (*n* = 90). (**B**) Ectopic expression of miR-122 suppresses LX-2 growth (*n* = 3). (**C**) The expression of fibrotic genes in the miR-122-tranfected LX-2 cells (*n* = 3). (**D**) Hydroxyproline (Hy:Pro) level in the CCl4-injected wild-type (WT) and miR-122 KO mice livers (*n* = 5). (**E**) Liver histology of the CCl4-treated WT and miR-122 KO mice (scale bar: 100 µm). (**F**) A schematic representation of the co-culture experiment involving LX-2 cells and control (empty vector-transfected) or miR-122 expression-vector-transfected HCC cells. (**G**) Mature miR-122 was measured by RT-qPCR in the purified extracellular vesicles from the co-cultured media shown in Figure 1F (*n* = 3). (**H**) RT-qPCR data in LX-2 cells co-cultured with the lentivirus-carrying empty vector-infected HCC (PLC/PRF5 (Vec-PLC and HCC-LM3 (Vec-LM3) or miR-122 expression-vector-infected HCC (PLC/PRF5 (122-PLC) and HCC-LM3 (122-LM3)) cell lines. Note: * *p* < 0.05, Student’s *t*-test.

**Figure 2 biology-09-00157-f002:**
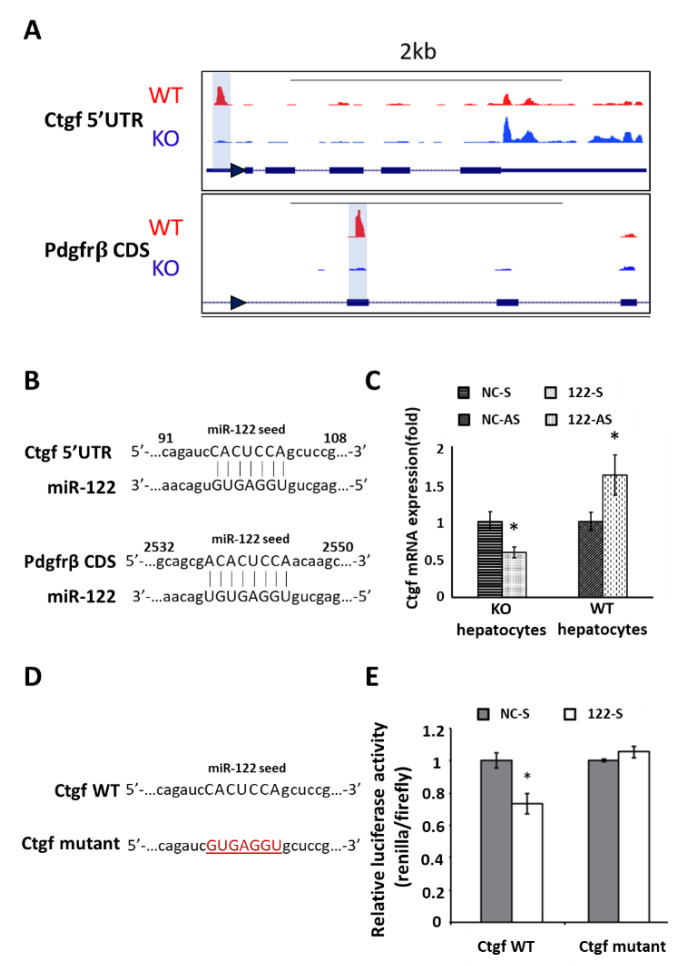
Argonaute-crosslinking immunoprecipitation (Ago-CLIP) analysis identified several pro-fibrotic targets of miR-122. (**A**) Examples of effects of Ago-CLIP-identified miR-122 seed sequence on its targets, Ctgf and Pdgfrβ. Note that the miR-122-dependent binding peaks are presented (highlighted in light blue) in the WT but not in KO liver. The arrows on the mRNA sequence indicate the directions of gene transcription. The solid and broken lines denote exons and introns, respectively. (**B**) The miR-122 binding site on the Ctgf 5′UTR and Pdgfrβ CDS. (**C**) Reciprocal expression of Ctgf and miR-122 in the primary KO mouse hepatocytes transfected with miR-122 mimic or NC mimic (KO hepatocytes) and WT hepatocytes transfected with anti-sense miR-122 or anti-sense NC (*n* = 3). NC-S: negative control for miR-122 sense; 122-S: miR-122 sense; NC-AS: negative control for miR-122 anti-sense; 122-AS: miR-122 anti-sense. (**D**,**E**) The miR-122 wild-type (WT) and mutant sequences in the 5′UTR of Ctgf cloned into psi-Check2 vector (**D**); each construct was co-transfected with either scramble RNA (NC-S) or miR-122 mimic (122-S) (*n* = 3) and luciferase (Renilla and firefly) activities were measured after 48 h (E). Data are presented as the fold change of the *Renilla* to firefly luciferase. Note: * *p* < 0.05, Student’s *t*-test.

**Figure 3 biology-09-00157-f003:**
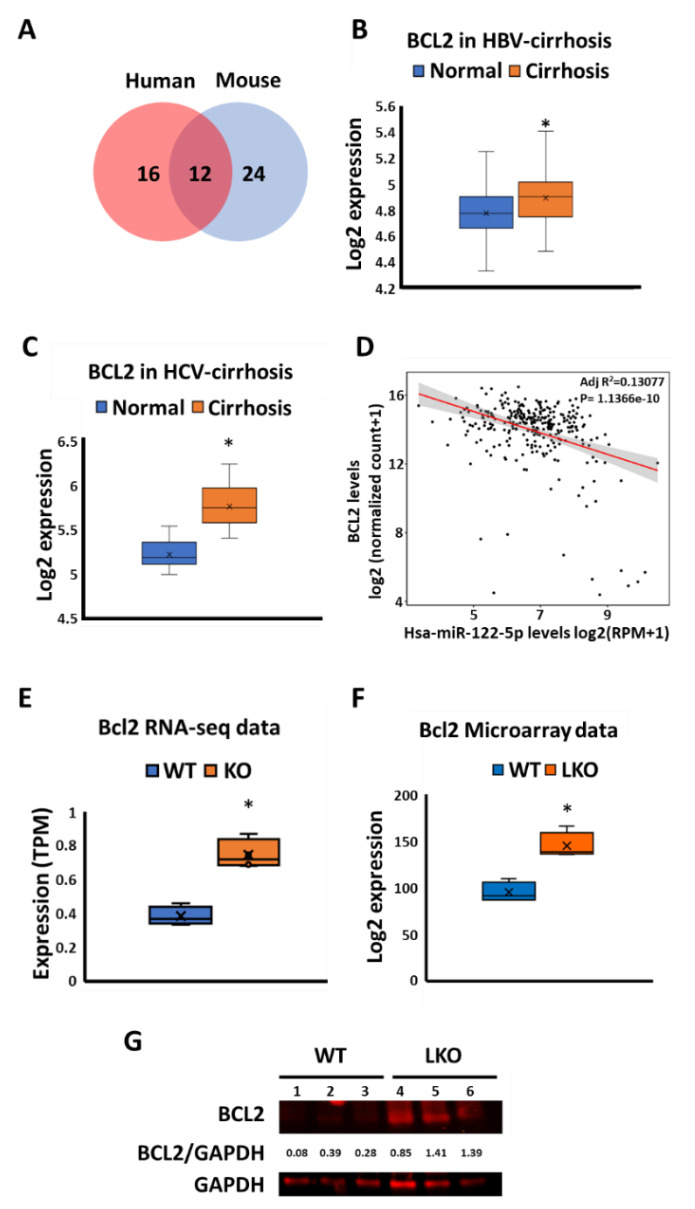
BCL2 is upregulated in both human and mouse fibrotic livers. (**A**) Overexpressed pro-fibrotic genes in the cirrhosis patients (red) overlay with those identified in the miR-122 KO mouse liver (blue). The *p*-value < = 0.001 was the cut off. (**B**) BCL2 expression in HBV-infection-induced cirrhotic liver tissues. Normal group: both inflammation and fibrosis scores = 0 (*n* = 26); cirrhosis group: fibrosis scores > = 1 (*n* = 81). (**C**) BCL2 expression in normal liver (*n* = 19) and HCV-infection-induced cirrhotic liver tissues (*n* = 41). (**D**) Correlations between miR-122 and BCL2 expression (Pearson’s correlation; intercept = 18.179; slope = -0.62463) in The Cancer Genome Atlas (TCGA) liver hepatocellular carcinoma (LIHC) with fibrotic livers (*n* = 292). (**E**) BCL2 expression determined from the RNA-seq data of wild-type (WT, *n* = 5) and miR-122 KO (*n* = 4) mice livers. (**F**) BCL2 expression determined from the microarray data of the wild-type (WT, *n* = 4) and miR-122 LKO (*n* = 4) mice livers. (**G**) BCL2 protein level in the liver lysates of WT (*n* = 3) and miR-122 liver-specific knockout (LKO) mouse (*n* = 3) at the age of 15 weeks. BCL2 levels were normalized to that of GAPDH and the value was indicated. Quantification was done by ImageJ (https://imagej.nih.gov/ij/). Note: * *p* < 0.05, Student’s *t*-test.

**Figure 4 biology-09-00157-f004:**
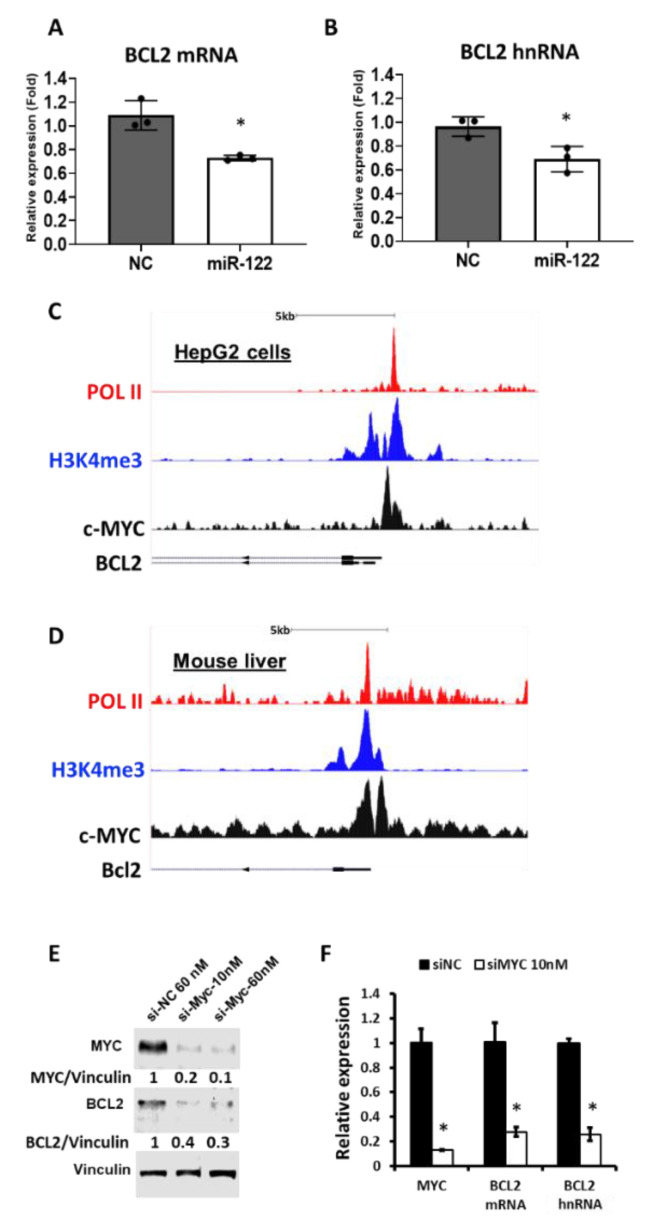
The miR-122 represses *BCL2* transcription through c-MYC. (**A**) The expression of *BCL2* mRNA (**A**) and hnRNA (**B**) in the miR-122-transfected LX-2 cells (*n* = 3). (**C**) ChIP-seq data of RNA polymerase II (POL II), H3K4me3, and c-MYC in the proximal promoter region of *BCL2* gene in the human HCC cell line HepG2 (accession numbers GSM822291 and GSM733737) and in the mouse liver (**D**) (accession number GSE76078). (**E**) BCL2 protein levels in the LX-2 cells transfected with either negative control RNA (Si-NC) or c-MYC siRNA (si-c-MYC). Vinculin was used as a loading control. (**F**) The expression of *c-MYC* mRNA, *BCL2* mRNA, and *BCL2* hnRNA (primary transcript) in the c-MYC siRNA-transfected LX-2 cells (*n* = 3). Note: * *p* < 0.05, Student’s *t*-test.

**Figure 5 biology-09-00157-f005:**
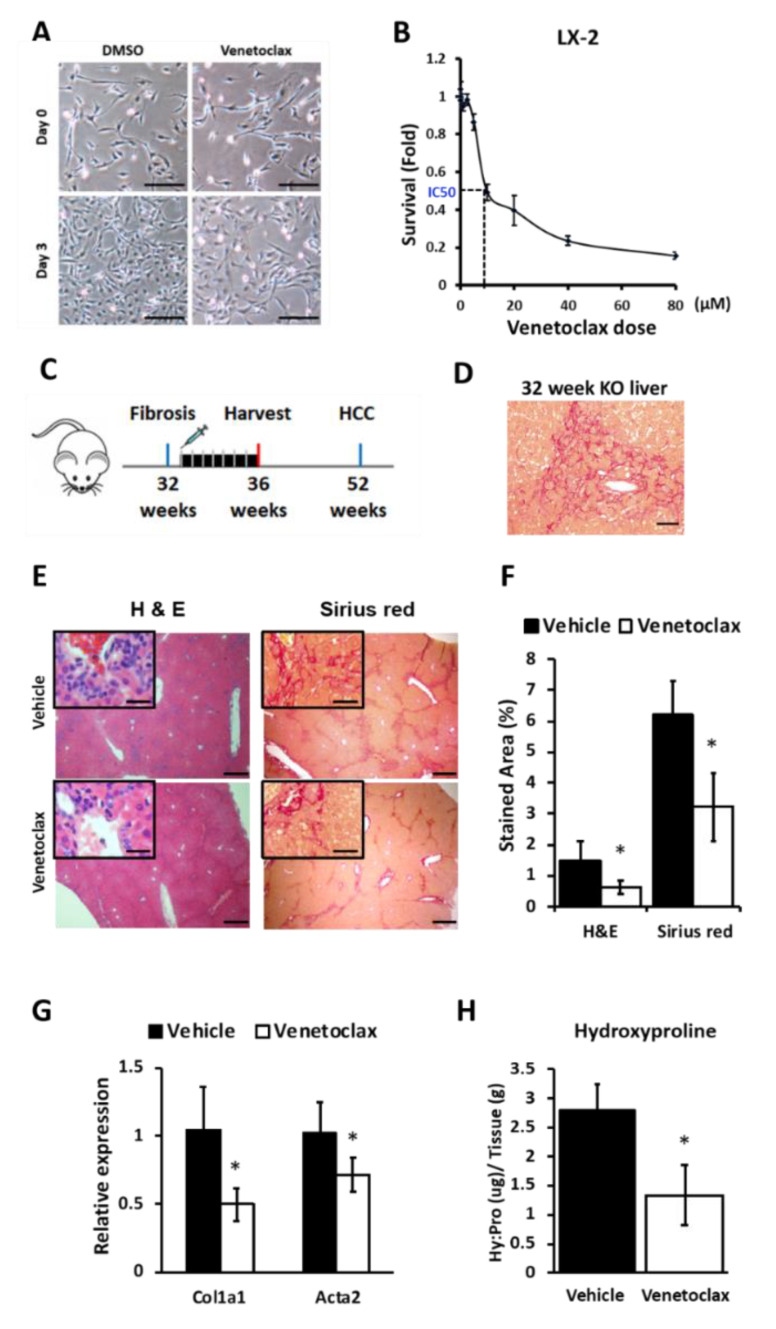
Venetoclax, a BCL2 selective inhibitor, suppresses hepatic stellate cell (HSC) activation. (**A**) Representative microscopic pictures (scale bar: 25 µm) of LX-2 cells treated with Venetoclax (10 µM) at 0 and 72 h (200× magnification). (**B**) Survival of LX-2 cell treated with different concentrations of Venetoclax for 72 h was measured by CellTiter-Glo^®^ Luminescent Cell Viability Assay (*n* = 3). Survival at 0 h was arbitrarily assigned a value of 1 and the survival in drug-treated cells compared to the vehicle-treated cells at each drug concentrations were plotted. (**C**) A schematic of Venetoclax treatment of miR-122 KO mice that spontaneously developed massive liver fibrosis at 32 weeks. (**D**) Representative Sirius red- stained pictures of 32-week-old KO liver (400× magnification; scale bar: 25 µm). (**E**) H&E and Sirius red staining of the vehicle and Venetoclax-treated mice liver sections (scale bar, 250 µm; inset, 25 µm). (**F**) Quantification of the H&E and Sirius red staining data in Figure 5E. Quantification was done by ImageJ (https://imagej.nih.gov/ij/) of three randomly chosen fields (100× magnification) per animal (*n* = 5). (**G**) *Col1a1* and *Acta2* mRNAs in the liver were measured by RT-qPCR (*n* = 5). (**H**) Hydroxyproline levels were measured in the vehicle and Venetoclax-treated mice liver tissue extracts. Hydroxyproline contents (µg) were normalized to the respective liver tissue weights (g) used in the assay (*n* = 5). Note: * *p* < 0.05, Student’s *t*-test.

**Figure 6 biology-09-00157-f006:**
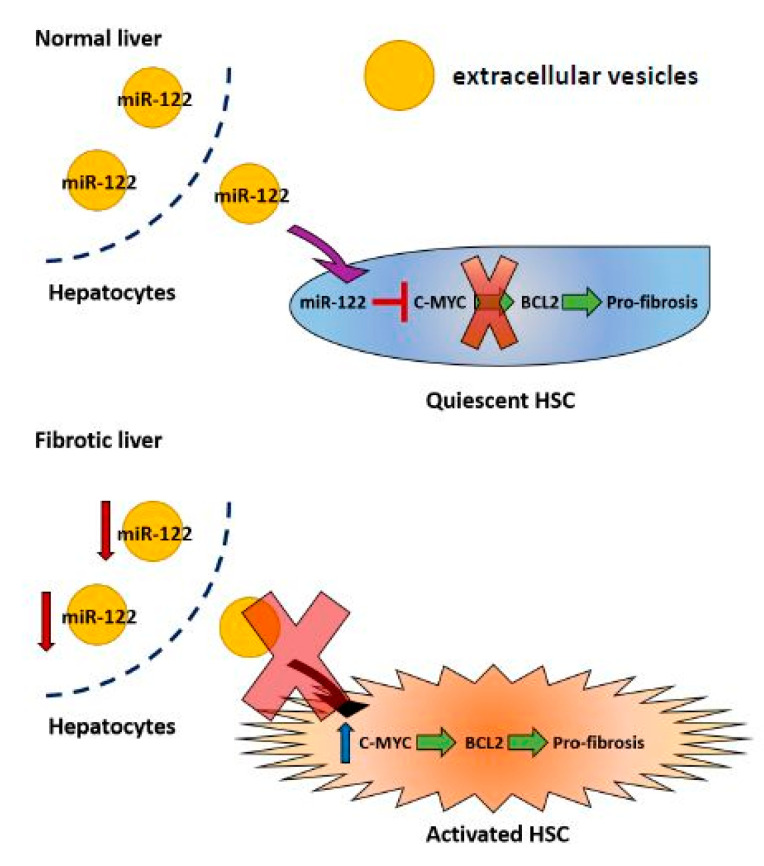
A model depicting the anti-fibrotic role of miR-122 in the liver. In healthy livers, miR-122 could be delivered to HSCs to inhibit the expression of fibrogenic BCL2 through suppressing c-MYC. When the miR-122 level goes down, c-MYC is de-repressed in the HSCs. Elevation of c-MYC enhances BCL2 transcription and increases the anti-apoptotic capability of HSCs.

**Table 1 biology-09-00157-t001:** Profibrotic targets of miR-122 identified by Ago-CLIP analysis of WT and miR-122 KO mice livers.

Gene	Annotation	Seed Type	*p*-Value	WT VS KO (log2)
Ctgf	5’UTR	7A1	0.016988	−1.518624491
Pdgfrb	CDS	8mer	3.75 × ^−21^	−0.96498004
Col5a1	CDS	8mer	3.26 × ^−20^	−0.952192613
P4ha1	3’UTR	8mer	0.001085	−3.232382621
Mmp2	3’UTR	7A1	1.02 × ^−24^	−1.307276327
Klf6	3’UTR	7m8	5.68 × ^−47^	−1.190961957
Col4a1	3’UTR	6mer	7.08 × ^−24^	−1.102863702
Vcam1	3’UTR	7A1	4.68 × ^−6^	−0.979919528
Ngfr	3’UTR	7m8	7.06 × ^−12^	−0.954727937
Myh9	3’UTR	6mer	1.67 × ^−16^	−0.796204606
Tnfrsf1b	3’UTR	6mer	0.002705	−0.766674249
Tgfbr1	3’UTR	7A1	3.83 × ^−6^	−0.332025716
Tgfbr1	3’UTR	6mer	3.83 × ^−6^	−0.332025716

**Table 2 biology-09-00157-t002:** Summary of the liver fibrosis score in the miR-122 knockout mice.

Treatment	Mouse ID	Score	Summary
Vehicle	5136	3	2.6 ± 0.55
	5137	3	
	5138	3	
	5139	2	
	5140	2	
Venetoclax	5141	2	1.2 ± 0.45
	5142	1	
	4963	1	
	4964	1	
	4190	1	

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
