# Peer review of "Role of B Cell Lymphoma 2 in the Regulation of Liver Fibrosis in miR-122 Knockout Mice"

_biology, 2020, doi:10.3390/biology9070157_

Round 1
Reviewer 1 Report
Teng et al. described a study of the role of B-cell lymphoma in the regulation of liver fibrosis in miR-122 knockout mice. The manuscript is missing all the figures; therefore, I cannot provide objective review. I cannot recommend publishing the current version of the manuscript in MDPI Biology.
Reviewer 2 Report
Title: Role of B cell lymphoma 2 in the regulation of liver fibrosis in miR-122 knockout mice
Summary: In this manuscript, the authors demonstrated role of miR-122 in liver fibrosis based on both in silico data and in vitro validation. Furthermore, regarding of mechanistic validation, they suggested Venetoclax, feasible Bcl2 inhibitor approved from FDA, as a therapeutic strategy to suppress liver fibrosis. It is a systemically organized study, but I have the following queries and comments.
Major comments;
- What is the regulation mechanism of miR-122. According to introduction, cytokines such as TGFb1, TNFa and IL6 from microenvironment, might activate quiescent HSC into myofibroblast like cell, after liver injury. In this context, after these kinds of cytokine exposure, dose miR-122 expression altered in hepatocyte?
- The authors demonstrated miR-122/Myc/Bcl2 axis to regulate liver fibrosis. Especially, they carefully examined regulation of Bcl2 expression by either miR122 or cMyc. However, the evidence that Myc-Bcl2 induce liver fibrosis is missing. Authors should clarify the notion that Myc dependent Bcl2 expression promotes liver fibrosis.
- In this research, they suggested venetoclax suppressed liver fibrosis. In previous researches, navitoclax is well studied as a senolytic agent, which could induce apoptosis in senescenced cell (PMID: 26657143) and cancer drug, which could lead to cell death selectively in mesenchymal like cell (PMID: 28606806).
3.1 Regarding of previous reports, does either bcl2 inhibition or venetoclax treatment induce apoptosis of liver fibrotic cell or suppress progression of liver fibrosis?
3.2 The effect of venetoclax in miR-122 expressing mice should be examined.
Minor comments;
- Fibrosis marker genes, such as aSMA and ATCA2 as well as miR122 need to be presented in Figure 1A
- Correlation between CTGF and fibrosis genes need to be examined in figure 2.
- In RNA-seq and microarray data, fibrosis genes depending on Bcl2 expression also need to be analyzed.
- typos need to be corrected
Reviewer 3 Report
The study of Teng et al. aims to explore the role of miR-122 in regulating liver fibrosis and identify downstream targets for treatment of liver fibrosis. Using public database and their own previously published or new data, they show 1) that mir-122 is under expressed in liver of patient with cirrhosis, that over expression of mir-122 in an human hepatic stellate cell line decreased proliferation and expression of fibrosis related genes and that mir-122 might be addressed to HSC from hepatocyte by exosome, 2) identification of potential targets of mir-122 using previously publish data, 3) CTGF as a first a potential target that could not be confirmed after inhibition in their mir-122 KO liver fibrosis model, 4) identification of BCL2 as a potential targets downstream of c-Myc that could be regulated by mir-122, 5) targeting Bcl2 represents a potential strategy to treat liver fibrosis. The manuscript is globally well written but will benefit of some modification to ease the read and understanding of some sections (see minor comments). The experiments and results describe yield interesting and potentially important data on the molecular events downstream mir-122 leading to liver fibrosis and the repurposing of Venetoclax for treatment of liver fibrosis. There are several figures and missing data that require clarification or could be improved to strengthen the conclusions of the manuscript and the proposed mir-122/c-Myc/Bcl2 axis in liver fibrosis. I have indicated below data/figures which require more data to support the conclusion.
Major comments
1) Fig1B: No control data are provided to show the extent of the overexpression of mir-122 in the LX-2 cells
2) Fig1G: No control data are provided to show the quality and purity of the exosome purification. Exosome preparation are often contaminated with other type of extra cellular vesicles or proteins. As microRNA could also be transported in present in combination with proteins, making sure of the purity of the exosome preparation will ensure that the expression of mir-122 is coming from the exosome
3) Fig1F-G-H: If the model of co-culture is an truly interesting approach to evaluate the paracrine delivery of mir-122 from hepatocytes to HSC, I have several issues with the interpretation and conclusion of this section:
a. A control condition with HSC culture in the bottom well without HCC in the transwell will help understanding if the presence of HSC impact the expression of Col1a1 and Acta2 directly and will give the opportunity to evaluate the extent of the effect of the HSC expressing mir-122 co-culture.
b. It would be of interest to show the effect of the co-culture on the proliferation of the HSC in the bottom well (including HSC with no HCC co-culture) to see if the system allows to achieve the same results as Fig1B
c. The inclusion of data on mir-122 on exosome is confusing and misleading as it suggests that mir-122 is delivered to the HSC by exosome from HCC. Several possibility could explain the effects observed in the co-culture system. 1) Mir-122 is delivered by another way from HCC to HSC 2) expression of mir-122 in HCC can change the secretory profile of the HCC and lead to secretion of cytokines or other factors that will operate on the HSC. As the authors already have in hand a protocol to purify exosome, I would recommend to treat directly the HSC with purified exosome to show or not the same effect as the co-culture system. If possible, showing the effect of exosome depleted conditioned media from HCC on HSCs will also greatly help evaluating and understanding the different causes of the co-culture results.
4) Fig2: If it is interesting to identify Ctgf as targets of mir-122, the fact that no details are provided about on the in-vivo model used to try to validate Ctgf targeting in-vivo and the lack of discussion on this result makes this figure not informative for the study. Please add more details about the strategy used for in-vivo CTGF targeting. Because of the role of CTGF in liver fibrosis depicted previously in several studies, it is warranted to develop a point of discussion concerning those results.
5) Fig3G: Please indicate age of mice for the measurement of BCL2 in liver. Is it before or after onset of fibrosis development?
6) Fig4: No data are depicted showing a direct effect on mir-122 on c-Myc level. If the Chip-seq data and SiRNA experiments are showing that c-MYC can control BCL2 the authors should provide direct evidence showing effect on mir-122 on c-MY expression on the LX-2 cells. The reference cited to support the mir-122 control of c-MYC level came from murine data, the authors have the opportunity here to show that it is also true in human HSC. Using the mir-122 overexpression condition depicted in Fig1B/Fig4A and measuring level of c-MYC at the protein level would be important to address this comment and support the model presented in Fig6. This is particularly important as c-MYC has been shown in other model to repress BCL2 expression.
7) FigF-G-H: Adding the value of the liver at 32weeks before treatment to both graphs will help understanding if the treatment by Veneoclax is stopping, slowing down or reverting the established liver fibrosis. This will strengthen the interest and potential repurposing of Venetoclax for liver fibrosis treatment.
Minor comments:
1) Line 179 to 182: Please rewrite this section to make it clearer
2) Line 211: “blocking the pro-fibrotic factor” replace by “blocking pro-fibrotic factors” are it is likely that not just only one pro-fibrotic factors can be targeted.
3) Line246 to 250: Please explain more clearly in the result section the ECl4 cell doxycycline system as it is not described in material and methods and will help understanding this nice piece of data supporting the regulation of BCL2 by c-MYC
4) As c-Myc overexpression has been shown in other studies to suppress BCL2, adding a discussion about the invert c-Myc effect on BCL2 expression would be interesting. As BCL2 has been shown to inhibit the pro-apoptotic effect of c-Myc this can emphasis the interest of BCL2 targeting by Venetoclax in liver fibrosis.
Reviewer 4 Report
The study performed by Kun-Yu Teng et al. presented a regulatory axis between miR-122 and BCL2, which, to some extent, explains the significance of miR-122 decrease in the pathogenesis of liver fibrosis. Before publication, there are several issues need to be addressed:
- The main figures are MISSING from the submitted documents!
- The link between BCL2 and fibrosis has been long realized. The main contribution of this study is the identification of the miR-122-MYC-BCL2 regulatory axis. However, the evidence is seemingly insufficient. First, the authors need to demonstrate if the MYC level is increased in fibrotic tissues compared with the normal tissues, and if there is any correlation between miR-122 and MYC level; Second, the interaction between MYC and BCL2 is controversial while most of the studies demonstrated that MYC, in fact, represses BCL2 expression. The binding of MYC to the promoter region does not direct mean activation, especially given that MYC can bind to a large proportion of genomic promoters. More data from both clinical sample analysis and in vitro functional study is required to prove the hypothesis.
Round 2
Reviewer 1 Report
Apart from the missing pictures, I had previously only minor comments, which were covered by other reviewers. I recommend publishing the manuscript in MDPI Biology as it is.
Reviewer 3 Report
Most of the suggested experiments haven't been properly addressed even for simple in-vitro assays.
Several un-conclusive points in different figures remain and should be addressed, especially the simple experiments not requiring development of novel technics or usage of new reagents. Below are the minimal set of data that should be added to support the conclusion and interest of this study.
Fig1G:
- The connection between vesicle and soluble factors in co-culture media was not addressed. This figure is at best suggestive and bring very little to the study as none of the other potential secreted factor were explored. As the authors already developed the purification process of those vesicle, it is at least expected to see data for which purified vesicles are directly added to HSC in media with exosome depleted FBS and impact the mRNA expression of the cells.
CTGF data:
- The discussion point of the authors on the relevance of the data for other team in the field is well taken and thus should be added as a point of discussion as previously recommended.
- As already asked, to be truly useful for the field, the authors should explain the strategy and condition used to block CTGF function in-vivo.
Fig4:
- The mRNA level of Myc after mir-122 overexpression gives a first line of evidence about miR-122 effect on Myc. This figure should be in the figure4 and not in supplementary data as it is a critical piece of data for the connection between miR-122 and Myc.
- Ultimately decrease of Myc protein level as asked in the first review is warranted in that case. Given the high level of over-expression of miR-122 in this system and the only slight decrease of Myc mRNA, showing Myc protein level would strengthen this crucial set of data as microRNA can act directly on protein translation. This is a simple experiment requiring to only have the cells in culture.
Several typo remains in the main text
Reviewer 4 Report
The authors have partially addressed the comments.